# Sensory Circumventricular Organs, Neuroendocrine Control, and Metabolic Regulation

**DOI:** 10.3390/metabo11080494

**Published:** 2021-07-29

**Authors:** Jin Kwon Jeong, Samantha A. Dow, Colin N. Young

**Affiliations:** Department of Pharmacology and Physiology, School of Medicine and Health Sciences, The George Washington University, 2300 I St NW, Washington, DC 20037, USA; jinkwon0911@gwu.edu (J.K.J.); sdow3@gwu.edu (S.A.D.)

**Keywords:** subfornical organ, organum vasculosum of the lamina terminalis, area postrema, hypothalamus, metabolism

## Abstract

The central nervous system is critical in metabolic regulation, and accumulating evidence points to a distributed network of brain regions involved in energy homeostasis. This is accomplished, in part, by integrating peripheral and central metabolic information and subsequently modulating neuroendocrine outputs through the paraventricular and supraoptic nucleus of the hypothalamus. However, these hypothalamic nuclei are generally protected by a blood-brain-barrier limiting their ability to directly sense circulating metabolic signals—pointing to possible involvement of upstream brain nuclei. In this regard, sensory circumventricular organs (CVOs), brain sites traditionally recognized in thirst/fluid and cardiovascular regulation, are emerging as potential sites through which circulating metabolic substances influence neuroendocrine control. The sensory CVOs, including the subfornical organ, organum vasculosum of the lamina terminalis, and area postrema, are located outside the blood-brain-barrier, possess cellular machinery to sense the metabolic interior milieu, and establish complex neural networks to hypothalamic neuroendocrine nuclei. Here, evidence for a potential role of sensory CVO-hypothalamic neuroendocrine networks in energy homeostasis is presented.

## 1. Introduction

Precise and reciprocal interactions between the central nervous system (CNS) and peripheral organs plays an integral role in whole body metabolic homeostasis, and impairments in this CNS-peripheral communication are clearly implicated in the development of metabolic disorders. This encompasses a wide range of conditions including obesity, type II diabetes, hypertriglyceridemia, non-alcoholic fatty liver disease, and insulin resistance, to name a few [1,2,3,4,5,6]. Within the CNS, a network of brain regions are involved in metabolic regulation, however, it is generally accepted that metabolic information from both peripheral and central inputs will eventually be integrated into the hypothalamus [4,5,7]. Hypothalamic nuclei, in particular the paraventricular nucleus (PVN) and supraoptic nucleus (SON), possess a wide array of neuroendocrine neurons, and therefore are considered as regions central to neuroendocrine regulation. However, the majority of circulating factors (hormones, adipokines, metabolites, etc.) cannot directly access these hypothalamic nuclei as they are protected by the blood-brain barrier (BBB) and/or substances are transported in limited quantity across the BBB; specialized endothelial cells located between the bloodstream and brain as a protective barrier against circulating toxins and pathogens [8,9]. This suggests involvement of other brain region(s) upstream of the PVN/SON in neuroendocrine-dependent metabolic homeostasis. In this regard, the sensory circumventricular organs (CVOs) are a key candidate, considering that: (1) They are located outside the BBB; (2) They possess the cellular machinery to detect circulating information, and; (3) They establish direct and/or indirect synaptic networks to hypothalamic neuroendocrine nuclei (Figure 1). Here, we will discuss existing anatomical, functional, and circuit level evidence pointing to the involvement of sensory CVOs in neuroendocrine regulated control of metabolism.

## 2. Arcuate Nucleus Involvement in Metabolic Regulation

Before discussing a neuroendocrine-metabolic role of the sensory CVOs, it is important to consider what has been the predominant focus of the field. Since the identification of dense leptin receptors in hypothalamic nuclei [10], numerous investigations have focused on hypothalamic neural circuits in whole body metabolic regulation, in particular an arcuate nucleus-dependent axis [3,11]. The arcuate nucleus is a small region located in the mediobasal hypothalamus adjacent to the third ventricle (3V) and median eminence. While some studies have proposed the arcuate nucleus as a part of the CVOs [12], this region in fact possesses an intact BBB, and is therefore, fully protected from the circulation [13]. Nevertheless, the arcuate nucleus plays a key role in metabolic regulation, due to an ability of circulating factors to access the region through the median eminence and/or median eminence-3V complex [14,15]. 

The arcuate possesses two functionally opposing neuronal populations: neurons expressing proopiomelanocortin (POMC) and those producing agouti-related peptide (AgRP) and neuropeptide Y (NPY) [3,16,17]. Although these neuronal populations synaptically innervate multiple brain regions, hypothalamic neuroendocrine nuclei, particularly the PVN, are main targets. Conversely, POMC and AgRP/NPY receive dense inputs from regions throughout the CNS (detailed in ref [18]). When activated by satiety signals, such as leptin, estrogen, and insulin, POMC neurons produce and release alpha-melanocyte stimulating hormone (α-MSH) into other brain regions (e.g., PVN) as a neurotransmitter to decrease appetite while also increasing energy expenditure [3,19]. On the other hand, AgRP/NPY neurons release hunger factor-induced inhibitory neurotransmitters to negatively regulate POMC neuronal activity (Jeong 2014). In brief, a balance between POMC and AgRP/NPY neurons is thought to be key to the modulation of energy homeostasis [16,17]. 

In addition to neuronal populations within the arcuate nucleus, astrocytes and tanycytes, specialized glial cells located on the bottom of the 3V wall, also express a broad array of metabolic receptors. Multiple investigations have suggested a role for these glial cells as a means to communicate and introduce circulating metabolic cues to arcuate neurons [6,15,20]. The arcuate nucleus also receives metabolic information from the gastrointestinal tract indirectly via brainstem nuclei [19]. Collectively, while the critical role of the arcuate in metabolic regulation is well established, it is important to consider distributed CNS networks that operate in concert or independently from the arcuate in metabolic regulation. In this context, emerging evidence points to a unique role of the sensory CVOs, as detailed below. 

## 3. Anatomy and Potential Metabolic Role of the Sensory CVOs

Most capillaries in the brain establish a BBB—a complex cellular physical barrier to protect the brain from the circulation [21]. While endothelial cells that are connected to each other through tight junctions are the basic component of the BBB, other neuronal and non-neuronal cells also form the BBB, which results in minimal fenestration and/or requires transport of select molecules [8,9]. However, the BBB in certain brain regions is “more loose” and permeable with discontinuous tight junctions, and therefore, blood-derived molecules can easily access the brain. These brain structures that lack a normal BBB are called the CVOs. The CVOs are comprised of secretory and sensory nuclei, of which the latter includes the subfornical organ (SFO), organum vasculosum of the lamina terminalis (OVLT), and area postrema (AP) [22,23]. Each of the sensory CVOs establishes neural networks, directly or indirectly, to the hypothalamus, and accumulating evidence suggests that signaling in the sensory CVOs may modulate broad metabolic parameters through hypothalamic control [23,24,25]. The unique characteristics and existing evidence that points to a neuroendocrine-dependent metabolic regulatory role of the SFO, OVLT, and AP is summarized below.

### 3.1. The SFO

The SFO is a sensory CVO located at the midline of the brain within the lateral ventricle and is comprised of two anatomically distinct subregions including the outer shell and ventromedial core [8,26,27,28]. Evidence suggests differential arrangement of tight-junction molecules within these SFO subregions, which impact size-dependent permeability of blood-borne molecules [8]. For example, peripheral administration of permeability indicators revealed that small molecules (<3000 kDa) accumulated primarily within the collagen IV-enriched ventromedial core, while the laminin-dominant outer shell was more selective for larger molecules (>10,000 kDa). Another example is that the hormone angiotensin-II (Ang-II) activates primarily the ventromedial core of the SFO, as represented by c-Fos expression following peripheral Ang-II administration, despite Ang-II type 1a receptors (AT_1a_R) being broadly distributed throughout the SFO [28]. However, it is still unclear whether anatomically distinct SFO subregions are responsible for differential physiological outputs. Instead, multiple cell phenotypes within the entire SFO have been demonstrated to play a pivotal role in metabolism regulation [29,30,31].

The SFO is well recognized for its role in cardiovascular and fluid balance regulation [32,33,34]. However, emerging evidence from transgenic reporter mouse models and transcriptomics also suggests a role in metabolic control due to a wide distribution of cellular receptors within the SFO, including receptors for insulin, leptin, estrogen, ghrelin, and adiponectin [1,35,36,37,38,39]. Furthermore, dynamic regulation of SFO metabolic receptors in response to fasting and overnutrition has been demonstrated with transcriptome analysis [38,39]. In addition, multiple electrophysiological investigations have also demonstrated the responsiveness of the SFO to multiple metabolic and inflammatory factors, such as leptin, amylin, ghrelin, and tumor necrosis factor-α [40,41,42,43,44,45]. Dynamic responsiveness of SFO cells to multiple metabolic factors is also evident [35,36,41,42,43,44,46,47,48]. For example, while some SFO neurons were activated in response to glucose, insulin, or adiponectin, other SFO neurons were either deactivated or non-responsive to the same stimulus [35,36,43,47]. SFO neuronal responsiveness to adiponectin has also been shown to be modulated by food deprivation [35]. These results indicate metabolic status-dependent, selective, and dynamic SFO cellular plasticity in response to metabolic substances.

While the aforementioned evidence collectively points to a role for the SFO in metabolic regulation, to date, in vivo evidence is rather limited. However, acute electrical stimulation of the SFO induces feeding in satiated animals [40], and peripheral administration of a synthetic melanocortin receptor agonist has been suggested to reduce overnight food intake in rats via the SFO [49]. These limited findings suggest possible involvement of the SFO in the regulation of feeding behavior, although future studies are clearly warranted. Moreover, hormonal signaling within the SFO may modulate whole body energy homeostasis independent of food intake. For example, selective removal of SFO insulin receptors in mice results in a metabolic syndrome-like phenotype accompanied by moderate elevations in body weight, adiposity, and the development of hepatic steatosis [1]. In addition, central administration of the adipokine leptin induces weight loss and upregulates sympathetically-mediated brown adipose tissue thermogenesis; responses that are dependent on SFO Ang-II signaling [29]. In line with this, multiple investigations have suggested possible involvement of the SFO in the development of metabolic disorders including obesity [30,50] and associated conditions such as non-alcoholic fatty liver disease [51]. For instance, neuroinflammation is strongly implicated in obesity development in rodents and humans [52], and investigations in rodents suggested involvement of SFO Ang-II signaling, at least in part, in high fat diet-induced neuroinflammation and obesity development [30]. Collectively, this emerging evidence points to a key role for the SFO in metabolic regulation, including potentially complex interactions between different hormones, although further work is necessitated.

### 3.2. The OVLT

Located at the rostral end of the third ventricle, the OVLT is a hypothalamic sensory CVO [53] that is divided by two anatomically and functionally independent subregions including the inner capillary plexus and outer lateral zone [8]. Small molecules in the circulation access the capillary plexus and then sequentially diffuse to the lateral zone within the OVLT; this phenomenon has been associated with heterogeneous expression of capillary tight junction molecules between the two OVLT subregions [8]. However, several anatomical studies have suggested that functional regulation by the OVLT may occur primarily in the lateral zone [8,28,54,55]. For example, both mRNA and protein levels of AT_1a_R were detected throughout the entire OVLT [56,57], but peripheral administration of Ang-II results in c-Fos expression predominantly within the lateral zone [28,54]. Additionally, astrocytes, which are critical for the sensing of circulating factors in this brain region [58,59,60] are primarily distributed in the lateral zone [8]. Even within the lateral zone, estrogen receptor-alpha (ERα) expression, a potential area where interactions between sex hormones and metabolic/cardiovascular/fluid information occurs, is exclusively clustered at the dorsal cap area [55]. Therefore, it is plausible that the inner capillary plexus is an entrance for circulating substances, and the outer lateral zone integrates this information to drive OVLT-mediated outputs to downstream regions including hypothalamic neuroendocrine nuclei.

Multiple anatomical and biochemical investigations have demonstrated receptors for insulin, leptin, Ang-II, endothelin, estrogen, oxytocin, arginine vasopressin (AVP), and relaxin 1/3 in the OVLT [1,55,56,61,62,63,64,65,66,67]. Responsiveness of the OVLT to these circulating factors is also evident. For example, application of AVP into primary OVLT cell culture medium evoked increases in intracellular calcium [63,64]. Additionally, c-Fos expression in the OVLT was elevated by intracerebroventricular administration of leptin in rats on a normal chow diet [68]. These findings, along with others [55,61,62], collectively suggest that the OVLT possesses the ability to monitor and respond to overall metabolic status. Interestingly, the cellular expression profile of these metabolic receptors is rather complex. For example, oxytocin and AVP V_1_ receptors are present in both neurons and glial cells [63]. Additionally, both OVLT neurons and glia are able to sense extracellular osmotic changes [59,69,70]. However, AVP V_2_ receptors and ERα have been suggested to be expressed solely on neurons [63,65], while the expression of endothelin receptor-1 and toll like receptor-4 (TLR-4) are predominantly on glial cells [58,62]. It is further possible that multiple metabolic factors may interact within the same OVLT cell. For example, the majority of OVLT ERα-expressing neurons (i.e., responsive to estrogen) are also osmosensitive, and dehydration-evoked hypertonicity induces c-Fos expression within ERα-expressing cells [55]. Therefore, OVLT-mediated metabolic regulation could be determined by complex intra-OVLT cellular interactions whereby circulating substances act upon similar and/or discrete cell types. 

In spite of the expression of numerous metabolic receptors in the OVLT, detailed in vivo investigations into OVLT-dependent metabolic regulation are currently lacking. This may be partially because the OVLT is a tiny structure located deep in the brain, and therefore, it is technically challenging to modify cell- and/or receptor-specific signaling pathways in this nucleus. However, several studies point to a potential role for the OVLT in energy homeostasis. For example, administration of the ovarian hormone relaxin peripherally or the neurohormone relaxin-3 directly into the brain induced OVLT neuronal activation and resulted in an increase in food intake in rats [67,71,72,73]. On the other hand, chemical blockade of the OVLT with acute administration of colchicine reduced food intake and blunted body weight gain [74]. In line with this, several investigations have also suggested OVLT involvement in food anticipatory behavior [75,76]. For example, in rabbit pups, increases in OVLT neuronal activity (i.e., c-Fos) was observed prior to scheduled nursing time [76]. It is also worthy to consider that the OVLT is well-recognized for its role in fluid balance. Metabolic and fluid regulation are closely related, and body fluid conditions can directly influence metabolic parameters, such as energy expenditure and food intake, both in humans and rodents [77,78,79,80,81]. Therefore, the OVLT may play a central role in whole body energy homeostasis by combining circulating fluid and metabolic information, although in-depth and targeted studies are clearly needed.

### 3.3. The AP

Similar to the SFO and OVLT, the AP possesses a specialized anatomy that allows it to monitor and regulate circulating factors, including those involved in metabolic function. Situated in the wall of the fourth ventricle, the AP is the most caudal sensory CVO and consists of three anatomically distinct areas: the perivascular, central, and lateral zones [8]. It has been suggested that the AP possesses a vascular portal system very similar to the neurohypophysis, connecting the vessels to the capillary plexus of the neuropil [82]. Sinusoidal vessels in the central zone of the AP, which is where most neurons and axon terminals reside, are much more fenestrated than the peripheral capillaries [83]. Thus, circulating molecules can directly access the central zone and then diffuse into the perivascular and lateral zones [8]. In line with this, glial cell bodies and fibers are dense in the lateral and perivascular zones, while the central zone shows very sparse glial immunoreactivity [8].

The majority of AP receptor expression is for hormones with anorexigenic effects, including amylin, CCK, GLP-1, peptide YY (PYY), adiponectin, and leptin. However, the AP is also equipped to detect orexigenic ghrelin [46,84]. Additionally, receptors for Ang-II, AVP, estrogen, and potentially insulin have also been identified in the AP [84,85,86,87,88]. Further characterization of receptor expression in specific cell types has revealed that amylin, leptin, Ang-II, GLP-1, adiponectin 1/2, CCK, and ghrelin receptors are expressed in AP neurons [89,90,91,92,93,94,95,96]. Leptin, TLR-4, glial-cell derived neurotrophic factor receptor α-like (GFRAL) and complement type 3 (a receptor linked to hypoxia-induced emesis) receptors are also localized on glial cells in this brain region [97,98,99,100]. mRNA expression of AVP V_1_a and PYY Y_1_ receptors have been detected in the AP; however, the specific AP cell types expressing these receptors is currently unclear [101,102].

A role for the AP in metabolic regulation is further supported by histological and electrophysiological findings demonstrating responsiveness to the administration of various anorexigenic hormones. Peripheral administration of amylin, CCK, GLP-1, PYY, and adiponectin all lead to increased c-Fos expression in AP neurons [103,104,105]. Furthermore, amylin, CCK, PYY, insulin, and adiponectin have all been found to influence the excitability of AP neurons. For example, the use of the α-amino-3-hydroxy-5-methyl-4-isoxazolepropionic acid receptor antagonist cyanquixaline to block amylin-induced excitatory responses revealed that administration of amylin excites AP neurons by facilitating glutamate release from glutamatergic inputs to AP neurons. Similar effects have been identified for CCK [94,106]. Interestingly, heterogenous responsiveness of AP neurons to metabolic factors has also been demonstrated. For example, in culture, low concentrations of PYY_1-36_ depolarize whereas high concentrations of PYY_3-36_ hyperpolarize AP neurons [84]. In addition, adiponectin and leptin primarily result in depolarization of most AP neurons. However, a subpopulation hyperpolarizes in response to these adipokines [46,90]. Smith et al. also demonstrated that the same subpopulation of AP neurons was responsive to both amylin and leptin, which was further supported by the demonstration that 94% of tested AP neurons were excited by both glucose and amylin [46,107]. In addition to anorexigenic hormones, the AP also appears to respond in a potentially complex manner to orexigenic peptides. Specifically, ghrelin induces hyperpolarization in 50% of AP neurons via modulation of voltage-gated K^+^ currents whereas the remaining ghrelin-sensitive neurons depolarize through a nonspecific cation current [108]. Collectively, these findings indicate that the AP is well-situated to integrate multiple circulating factors and responds to anorexigenic/orexigenic hormones, glucose, and adipokines, although the intricacies of the AP’s responsiveness to metabolic factors warrant further interrogation.

In line with the aforementioned receptor expression, and the well-recognized role of the AP as a “chemoreceptor trigger zone” due to its role in emesis [109], numerous studies have demonstrated AP activation following peripheral injection of various hormones. Peripheral administration of anorexigenic hormones amylin, CCK, GLP-1, PYY, and adiponectin all lead to increased c-Fos expression in AP neurons [103,104,105]. Furthermore, these hormones suppress feeding behavior in rodents, and this effect requires an intact AP and receptor activation [110,111,112]. For example, AP-specific blockade with the amylin receptor antagonist AC187 inhibited amylin-induced-feeding suppression, as well as feeding-induced c-fos expression in fasted rats. [103,110]. A role the AP in response to “newer” anorexigenic factors is also emerging. Specifically, growth differentiation factor 15 (GDF15), a stress response cytokine that signals via GFRAL, inhibits feeding [113]. Activation of GFRAL receptors induce AP neuron activation, suggesting that GDF15-induced suppressed food intake may be mediated by the AP [114,115]. Hormones at the AP also influence other metabolic outcomes including thermogenesis and glucose homeostasis. For example, retrograde tracing from interscapular brown adipose tissue has implicated the AP in brown adipose tissue thermogenesis [116]. Additionally, mice with knockout of certain amylin receptor subunits become glucose intolerant [117]. Glucose intolerance also occurs in GFRAL knockout mice challenged by high-fat diet, which may be mediated directly by the AP or indirectly through the adjacent nucleus tractus solitarius (NTS) [118]. Together, these in vivo findings support the AP’s role in integrating circulating metabolic factors to regulate various physiological outcomes, potentially through direct AP-hypothalamic pathways or indirectly through AP-brainstem-hypothalamic neural pathways [114,118] as discussed below.

## 4. Sensory CVOs and Hypothalamic Circuits in Metabolic Regulation

As described above, the sensory CVOs are equipped with an array of receptors and responsive to numerous stimuli, making them a key entry point for circulating metabolic factors to influence the brain. Once detected and integrated in the sensory CVOs, this information will then be transmitted via neuronal efferents to hypothalamic metabolic centers, including the PVN and SON. Evidence for sensory CVOs-hypothalamic neuroendocrine neural networks is discussed below. 

SFO neurons establish direct as well as indirect synaptic connectivity with hypothalamic metabolic nuclei. For example, the SFO provides direct excitatory synaptic inputs to the PVN and SON [119,120,121]. In particular, cells within the dorsolateral peripheral subregion of the SFO project to the magnocellular portion of the PVN where numerous AVP and oxytocin cells are distributed [121]. The SFO also establishes excitatory and inhibitory synaptic communication with other hypothalamic nuclei, including the bed nucleus of the stria terminalis, arcuate nucleus, OVLT, and median preoptic nucleus (MnPO) [119,120,122,123,124]—neuronal networks that also allow the SFO to communicate indirectly with the PVN and SON. Interestingly, the cellular and synaptic architecture from the SFO to hypothalamus is very complex. For example, separate populations of SFO neurons project to the PVN and MnPO, although a weak number of SFO neurons provide collateral projections to both regions [125]. Importantly, the SFO and MnPO establish reciprocal connections, and SFO cells that receive inputs from the MnPO project to the PVN [126], suggesting a possible feedback loop between the SFO and MnPO to regulate an SFO-PVN axis. However, anatomical and synaptic projection information for specific SFO cell types, particularly in the context of “metabolic receptor” expressing neurons, is largely unavailable, and therefore needs to be addressed in the future.

Similar to SFO, OVLT-dependent metabolic regulation is most likely mediated by complex OVLT neural networks to multiple hypothalamic nuclei. However, in depth investigations are lacking, particularly as related to traditional metabolic mediators (e.g., adipokines, hepatokines, insulin, etc.). Nevertheless, insights from other areas of investigation provide insight into potential OVLT networks. In the context of thirst control and drinking behavior, the OVLT provides both excitatory and inhibitory inputs to the MnPO [124], and this information is further transmitted to the PVN [127]. Similarly, OVLT neurons expressing ERα, relaxin, AT_1a_R, and cholinergic receptors are also connected to the PVN and SON, presumably through the MnPO [28,65,71,128,129]. On the other hand, OVLT neurons that respond to extracellular sodium concentrations establish monosynaptic projections to the PVN [59,130]. Although indirect, given that fluid balance and metabolic regulation are closely related in humans as well as non-human species [77,78,79,80,81], these findings point to possible OVLT-hypothalamic networks that may be involved in metabolism regulation. 

Anatomical studies using retrograde tracers indicate that the AP sends efferent projections to the PVN and SON [131,132]. In line with this, hypertonic saline induces c-Fos expression in the PVN via the AP [133], indicating the existence of direct synaptic communications between the AP and hypothalamic neuroendocrine centers. However, more evidence is required to delineate the direct networks between the AP and PVN/SON. Nevertheless, the AP establishes strong bidirectional synaptic interactions with adjacent nuclei, including the NTS and dorsal motor nucleus (DMN). Numerous studies have suggested this AP-NTS-DMN cluster as a critical brainstem metabolic center [113,132,134,135,136,137,138,139]. Importantly, this brainstem metabolic complex is highly connected to hypothalamic neuroendocrine centers [140,141,142,143]. Thus, similar to the SFO and OVLT, the AP is anatomically situated to directly and/or indirectly influence metabolic regulation through hypothalamic neuroendocrine nuclei.

Within the hypothalamus, numerous neuroendocrine neuron subpopulations are distributed in the PVN and SON. To date, direct anatomical evidence into the precise neuroendocrine neuron type that the sensory CVOs project to remains uninvestigated. However, indirect evidence points to hypothalamic AVP and/or oxytocin neurons as a common downstream target of the sensory CVOs. For example, several hormones that are involved in metabolic regulation, including estrogen, relaxin, and Ang-II, have been shown to modulate gene expression and release of AVP and oxytocin through the sensory CVOs [65,71,144,145,146,147,148]. Additionally, SFO-specific electrical stimulation resulted in elevations in circulating AVP and oxytocin [149,150]. In line with this, pharmacological cholinergic stimulation of the SFO induced elevations in c-Fos expression within AVP cells in the PVN and SON [151]. Similarly, the OVLT, particularly OVLT neurons that directly project to the PVN, have been suggested to play a role in hyperosmolality-dependent AVP and oxytocin release [130,152,153]. In addition, relaxin administration in rodents also induces c-Fos expression in the PVN and SON that is paralleled by release of AVP and oxytocin; a response that is, at least in part, through OVLT mechanisms [71,146]. Peripheral administration of anorexigenic CCK induced SON oxytocin neuronal activity, and further, release of oxytocin into the bloodstream, which was blunted following AP lesioning [154]. Furthermore, central administration of GLP-1 increases plasma AVP levels, which is accompanied parallel increases in c-Fos in the AP, PVN, and SON [155].

The findings pointing to a sensory CVO influence on hypothalamic AVP and oxytocin neurons is intriguing given oxytocin and AVP’s ability to modulate a variety of metabolic outcomes including feeding behavior, body composition, and glucose/lipid metabolism. Oxytocin has been shown to exhibit anorectic effects, as both central and peripheral oxytocin administration leads to decreased food intake in animal models and humans [124,156,157,158,159,160,161]. Not only does oxytocin influence feeding behavior, but it further affects body composition and energy expenditure. In multiple animal models, loss of central oxytocin signaling via oxytocin neuron ablation or oxytocin receptor deletion increases fat mass and decreases energy expenditure [162,163,164,165]. Furthermore, recent work suggests that exogenous oxytocin treatment is associated with increased brown adipose tissue thermogenesis and “browning” of white adipose tissue, which is consistent with the increased energy expenditure induced by oxytocin treatment [166,167,168]. Changes in body composition may be further attributed to oxytocin modulation of glucose and lipid metabolism. Oxytocin enhances glucose uptake in muscle and adipose tissue and augments lipolysis and β-oxidation in adipose tissue [169,170,171,172]. 

Similar to oxytocin, AVP also affects a broad spectrum of metabolic parameters [173]. For example, acute endogenous activation of PVN AVP neurons decreases food intake, and peripheral administration of AVP further decreases brown adipose tissue thermogenesis in healthy rodent models [174,175,176]. On the other hand, hypothalamic AVP expression in rats is also increased with the onset of diabetes mellitus [177], suggesting a normal and pathophysiological effect of AVP in metabolism regulation. Interestingly, while AVP V_1_a receptor-deficient mice display enhanced hepatic glucose production accompanied by high plasma glucose levels [178,179], AVP V_1b_ receptor-deficient animals develop hypoglycemia [180], indicating AVP involvement in glucose homeostasis in a receptor-dependent manner. AVP also appears to prevent lipolysis and β-oxidation via V1a, as V1a-deficient mice display enhanced lipolysis in brown adipocytes and β-oxidation in muscle and liver [181]. In humans, the metabolic effects of AVP are unclear, however, several investigations have also suggested a link between AVP and metabolic disorders, such as obesity and diabetes [182,183,184]. 

## 5. Conclusions

It is well-accepted that hypothalamic neuroendocrine nuclei including the PVN and SON play a central role in regulating energy homeostasis. While the predominant and well-accepted focus has been on the role of arcuate nucleus influence to these regions, emerging results further suggest the involvement of non-hypothalamic brain regions including the sensory CVOs. Each of the sensory CVOs establishes direct as well as indirect synaptic communication with the PVN and SON. In addition, the sensory CVOs are located outside of the BBB and express a broad array of metabolic receptors. Therefore, the sensory CVOs are anatomically and biochemically situated to detect metabolic factors in the circulation and influence whole body energy homeostasis through downstream hypothalamic nuclei. While precise neuroendocrine modulation by the sensory CVOs continues to emerge, accumulating evidence points to AVP and oxytocin as potential neuroendocrine targets of the sensory CVOs in metabolic regulation. However, in-depth neuroanatomical and functional in vivo investigations are warranted to build upon existing work. Nevertheless, the sensory CVOs are likely brain sites that are involved in neural responses to circulating metabolic signals and play a key role in the central regulation of energy homeostasis through neuroendocrine mechanisms. 

## Figures and Tables

**Figure 1 metabolites-11-00494-f001:**
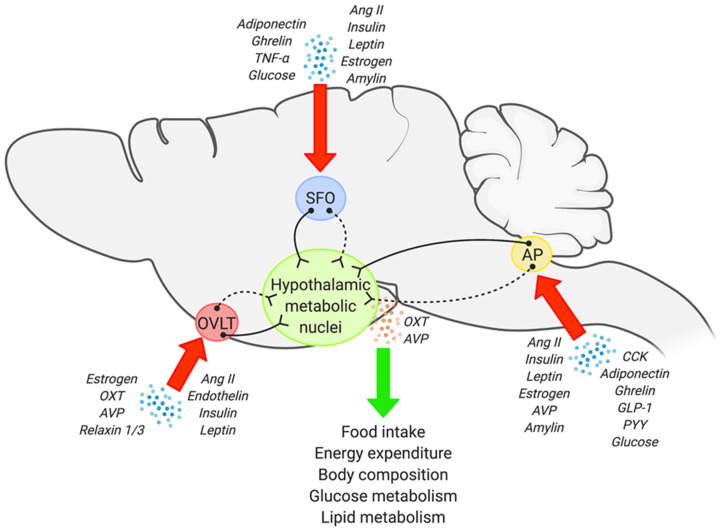
Schematic illustration showing potential sensory CVO-hypothalamic networks involved in metabolism regulation. Each of the sensory CVOs possesses the cellular machinery to sense multiple metabolic factors, a few of which are shown in the image. At the same time, sensory CVOs also establish direct (solid line) as well as indirect synapses (dashed line) to hypothalamic metabolic nuclei including the PVN and SON. Multiple investigations have demonstrated the involvement of the sensory CVOs in metabolism regulation, and further suggest that hypothalamic AVP and oxytocin (OXT) may play a key role. Image was created with Biorender.com.

## Data Availability

Not applicable.

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
