# Peer review of "Sensory Circumventricular Organs, Neuroendocrine Control, and Metabolic Regulation"

_metabolites, 2021, doi:10.3390/metabo11080494_

Round 1

Reviewer 1 Report

Dear authors,

I found your manucript to be well-researched and well-written.

A minor comment - please double check the abbreviations throughout the text.

Great figure!

Author Response

I found your manuscript to be well-researched and well-written.

A minor comment - please double check the abbreviations throughout the text.

Great figure!

We appreciate the reviewer’s valuable time and support of the manuscript. We have double-checked the abbreviations throughout the text.

Reviewer 2 Report

I suggest to improve the description of arcuate nucleus of hypothalamus.

I also suggest to mention main neuropeptides and neurotransmitters involved in transducing peripheral afferents for regulating energy balance. This could be done in paragraph #3.

Please use synthetic keywords.

Author Response

I suggest to improve the description of arcuate nucleus of hypothalamus. I also suggest to mention main neuropeptides and neurotransmitters involved in transducing peripheral afferents for regulating energy balance. This could be done in paragraph #3.

We appreciate the reviewer’s critical interpretation and suggestions. Based on your comments and those of Reviewer #3, we have included a new section in the revised manuscript providing an overview of the arcuate nucleus (Page 2, lines 48-82). This also includes discussion on neuropeptides/neurotransmitters involved in arcuate nucleus related control of energy balance. Importantly, in order to maintain the overall manuscript focus on the sensory circumventricular organs and also keep within the journal word limit, we have kept this section fairly broad. However, we agree that it is important to introduce what has been the predominant focus of the metabolic control field to the readers.

Please use synthetic keywords.

We apologize, but we were slightly confused by what the reviewer meant with this comment. We believe the keywords included in the manuscript (subfornical organ, organum vasculosum of the lamina terminalis, area postrema, hypothalamus, metabolism) encompass the broad focus of the manuscript without duplicating words in the article title as is common for NCBI listed publications. However, we are more than willing to add, remove, or modify any of the keywords if the reviewer feels it is necessary.

Reviewer 3 Report

In this review manuscript titled “Sensory circumventricular organs, neuroendocrine control, and metabolic regulation”, Jeong and collaborators summarized the literature that supports the concept that the sensory circumventricular organs (CVOs) are involved in the integration of responses mediated by the paraventricular nucleus (PVN) and the supraoptic nucleus (SON) to various types of metabolic information coming from the periphery. The authors postulate that since the PVN and SON are protected by the blood brain barrier (BBB), direct interacion of neuroendocrine neurons in these nuclei with circulating metabolites and hormones is unlikely. However, structures such as the subfornical organ, organum vasculosum of the lamina terminalis, and area postrema are potential candidates as points of detection or sensing of circulating substances because they are “located outside the blood-brain-barrier, possess cellular machinery to sense the metabolic interior milieu, and establish complex neural networks to hypothalamic neuroendo-crine nuclei.” Overall, the topic is of current interest, the manuscript is well written and nicely organized. However, the manuscript is biased towards the role of the CVOs and the PVN and SON leaving out of the picture other well established mechanisms (role of the arcuate nucleus, ARC). Even if the later is not the focus of the study, they should place their work in context with the extensive compelling evidence indicating the key role played by the ARC and other nuclei. Thus, there are certain questions that the authors would need to address to improve the scientific value of their manuscript.

Main issues:

1. Although the authors indicate that their manuscript focuses on the role of the PVN and SON, a large body of evidence indicates that the arcuate nucleus of the hypothalamus (a region often considered not to be protected by the BBB) plays a crucial role in the regulation of metabolism in general, including energy metabolism. In this regard, in the authors’ opinion, what would be the overall contribution of the CVOs to metabolic regulation. This is relevant considered the connections between neurons of the arcuate nucleus with other hypothalamic nuclei including the PVN. I think that the authors should put their hypothesis in perspective with what has already been established in this field and perhaps this should be incorporated into Figure 1.

2. Despite that circumstantial or sometimes more direct evidence points to a possible role of the CVOs in metabolic regulation, what the authors think about the fact that the contribution of the CVOs has not been more thoroughly investigated.

3. In the conclusions sections the authors state that “the sensory CVOs are likely brain sites that initiate neural cascades in response to circulating metabolic signals and play a key role in the central regulation of energy homeostasis through neuroendocrine mechanisms.” Please, clearly specify what specific pieces of evidence support this notion (for example, direct in vivo evidence).

Author Response

In this review manuscript titled “Sensory circumventricular organs, neuroendocrine control, and metabolic regulation”, Jeong and collaborators summarized the literature that supports the concept that the sensory circumventricular organs (CVOs) are involved in the integration of responses mediated by the paraventricular nucleus (PVN) and the supraoptic nucleus (SON) to various types of metabolic information coming from the periphery. The authors postulate that since the PVN and SON are protected by the blood brain barrier (BBB), direct interaction of neuroendocrine neurons in these nuclei with circulating metabolites and hormones is unlikely. However, structures such as the subfornical organ, organum vasculosum of the lamina terminalis, and area postrema are potential candidates as points of detection or sensing of circulating substances because they are “located outside the blood-brain-barrier, possess cellular machinery to sense the metabolic interior milieu, and establish complex neural networks to hypothalamic neuroendocrine nuclei.” Overall, the topic is of current interest, the manuscript is well written and nicely organized.

We thank the reviewer for your time, support, and comprehensive review of the manuscript.

However, the manuscript is biased towards the role of the CVOs and the PVN and SON leaving out of the picture other well established mechanisms (role of the arcuate nucleus, ARC). Even if the later is not the focus of the study, they should place their work in context with the extensive compelling evidence indicating the key role played by the ARC and other nuclei. Thus, there are certain questions that the authors would need to address to improve the scientific value of their manuscript. Although the authors indicate that their manuscript focuses on the role of the PVN and SON, a large body of evidence indicates that the arcuate nucleus of the hypothalamus (a region often considered not to be protected by the BBB) plays a crucial role in the regulation of metabolism in general, including energy metabolism. In this regard, in the authors’ opinion, what would be the overall contribution of the CVOs to metabolic regulation. This is relevant considered the connections between neurons of the arcuate nucleus with other hypothalamic nuclei including the PVN. I think that the authors should put their hypothesis in perspective with what has already been established in this field and perhaps this should be incorporated into Figure 1.

Based on your comments and those of Reviewer #2, we have included a new section in the revised manuscript providing an overview of the arcuate nucleus (Page 2, lines 48-82). This also includes discussion on neuropeptides/neurotransmitters involved in arcuate nucleus related control of energy balance. Importantly, in order to maintain the overall manuscript focus on the sensory circumventricular organs and also keep within the journal word limit, we have kept this section broadly focused. However, we agree that it is important to introduce what has been the predominant focus of the metabolic control field to the readers. In addition, it is important to note that we are not excluding the documented metabolic role of other brain regions including the arcuate nucleus. Importantly, we believe that discussion of a role for sensory circumventricular organs in metabolic regulation is in line with a growing body of literature indicating distributed brain networks involved in metabolic control, beyond the arcuate nucleus.

 Despite that circumstantial or sometimes more direct evidence points to a possible role of the CVOs in metabolic regulation, what the authors think about the fact that the contribution of the CVOs has not been more thoroughly investigated.

We agree with the reviewer; the contribution of the sensory CVOs in metabolic control continues to be an emerging area. However, this is the exact purpose of this manuscript – to indicate that there is accumulating evidence, whether circumstantial, indirect, or direct, - that these understudied brain areas may be critical regions for metabolic regulation. We don’t believe that the lack of current investigations represents a flaw in the current manuscript. Instead, this is an actual strength, in that the manuscript introduces an area to the field (just as was done for the arcuate and brainstem years ago) to allow for critical interpretation and further investigation.

In the conclusions sections the authors state that “the sensory CVOs are likely brain sites that initiate neural cascades in response to circulating metabolic signals and play a key role in the central regulation of energy homeostasis through neuroendocrine mechanisms.” Please, clearly specify what specific pieces of evidence support this notion (for example, direct in vivo evidence).

We apologize if the wording of this sentence came across as too direct and we have revised it to read: “The sensory CVOs are likely brain sites that are involved in neural responses to circulating metabolic signals and play a key role in the central regulation of energy homeostasis through neuroendocrine mechanisms.” (lines 392-394). We believe this statement is supported throughout the manuscript by the accumulating evidence pointing to the sensory CVOs – including molecular, anatomical, in vitro, and in vivo findings. We have highlighted numerous in vivo examples indicating participation of the sensory CVOs in metabolic regulation including, but not limited to, SFO insulin and leptin receptor regulation, OVLT involvement in body weight and food intake behavior, and AP participation in glucose and thermogenic regulation. Importantly, as indicated above, we are not implying that the sensory CVOs are the sole brain regions involved in metabolic regulation, but instead are part of a distributed and complex network that needs to be considered when evaluating neuroendocrine control of metabolism.  

Reviewer 4 Report

This review paper has made a good work on the potential role of the sensory circumventricular organs (CVOs)-hypothalamic neuroendocrine networks in energy homeostasis. The presented evidence indicates that each of 3 sensory CVOs (subfornical organ, organum vasculosum of the lamina terminalis, and area postrema) possesses the cellular machinery to sense multiple metabolic factors such as angiotensin II, insulin, leptin, etc. and sends the signal via direct and indirect synapses to hypothalamic metabolic nuclei including the paraventricular nucleus (PVN) and supraoptic nucleus (SON). While precise neuroendocrine modulation by the sensory CVOs continues to emerge, accumulating evidence suggests arginine vasopressin (AVP) and oxytocin as potential neuroendocrine targets of the sensory CVOs in metabolic regulation. There are some concerns as listed in the following:

(1) A recent review paper concerning direct and indirect action of AVP on metabolism (Peptides. 2021 Apr 24;142:170555. doi: 10.1016/j.peptides.2021.170555. Online ahead of print) can be cited.

(2) Typos and others

L11-12: the paraventricular nucleus of the hypothalamus and supraoptic nucleus.

L144: AVP V1 receptors

L238: BAT thermogenesis [105]

L305: elevations in AVP c-Fos

(3) It is better to keep one format for the paper title (capital letter only on the first word) in the References list: R4, R39, R43, R48, R57, R59, R67, R69, R85, R113, R129, R171, R172

Author Response

This review paper has made a good work on the potential role of the sensory circumventricular organs (CVOs)-hypothalamic neuroendocrine networks in energy homeostasis. The presented evidence indicates that each of 3 sensory CVOs (subfornical organ, organum vasculosum of the lamina terminalis, and area postrema) possesses the cellular machinery to sense multiple metabolic factors such as angiotensin II, insulin, leptin, etc. and sends the signal via direct and indirect synapses to hypothalamic metabolic nuclei including the paraventricular nucleus (PVN) and supraoptic nucleus (SON). While precise neuroendocrine modulation by the sensory CVOs continues to emerge, accumulating evidence suggests arginine vasopressin (AVP) and oxytocin as potential neuroendocrine targets of the sensory CVOs in metabolic regulation. There are some concerns as listed in the following:

We thank you for your support of the manuscript and careful review. We have incorporated your comments into the manuscript as indicated below.

A recent review paper concerning direct and indirect action of AVP on metabolism (Peptides. 2021 Apr 24;142:170555. doi: 10.1016/j.peptides.2021.170555. Online ahead of print) can be cited.

Thank you for this suggestion, and the suggested reference is now included in the text (line 364).

Typos and others

L11-12: the paraventricular nucleus of the hypothalamus and supraoptic nucleus.

L144: AVP V1 receptors

L238: BAT thermogenesis [105]

L305: elevations in AVP c-Fos

In the revised manuscript, we have corrected typos and abbreviations throughout (lines 11, 229-230, 271, 339).

It is better to keep one format for the paper title (capital letter only on the first word) in the References list: R4, R39, R43, R48, R57, R59, R67, R69, R85, R113, R129, R171, R172.

We apologize for these errors, which were a reflection of the citation software used. The reference list has been corrected in the revised manuscript (lines 426, 451, 456, 459, 466-467, 540-541, 552, 566-567, 589-590, 594-595, 616-617, 621-622, 662-663, 733-734, 773-774, 891-892, 894-895).

Reviewer 5 Report

In this work, the authors sought to describe the involvement of circumventricular sensory organs (CVOs) in neuroendocrine control and metabolic regulation. The subfornical organ (SFO), the organum vasculosum of the lamina terminalis (OVLT) and the postrema area (AP) establish, directly or indirectly, neural networks to the hypothalamus. These anatomical areas could modulate various metabolic parameters through hypothalamic control. Unfortunately the work cannot be accepted. The manuscript is unclear, in some parts even confused. The insertion of figures and tables could help the reader in understanding the text. The authors have reviewed numerous articles but perhaps, due to the large amount of information, they have not been able to better structure the work. Finally, authors should try to add personal reflections after discussing the data. I think the card should be refined and reorganized

Author Response

In this work, the authors sought to describe the involvement of circumventricular sensory organs (CVOs) in neuroendocrine control and metabolic regulation. The subfornical organ (SFO), the organum vasculosum of the lamina terminalis (OVLT) and the postrema area (AP) establish, directly or indirectly, neural networks to the hypothalamus. These anatomical areas could modulate various metabolic parameters through hypothalamic control. Unfortunately the work cannot be accepted. The manuscript is unclear, in some parts even confused. The insertion of figures and tables could help the reader in understanding the text. The authors have reviewed numerous articles but perhaps, due to the large amount of information, they have not been able to better structure the work. Finally, authors should try to add personal reflections after discussing the data. I think the card should be refined and reorganized.

We find it unfortunate that you found the manuscript unclear, confusing, and unstructured at times. We believe that this review article follows a logical flow and presents to the readership an overview of understudied yet emerging brain regions that may be involved in neuroendocrine control of metabolism. Indeed, Reviewers #1-4 all recognized merit in the manuscript and found it “well-researched and well-written,” “of current interest,” and “nicely organized.” We have included a number of revisions according to other comments raised in order to integrate the current manuscript with accepted tenets in the field. At the current time, we have not included personal reflections at the end of each section. Instead, we have summarized the current literature in an integrative manner to allow the reader to arrive at their own conclusions. We do not feel that it is appropriate to point the reader in a specific direction, but instead allow them to synthesize what we have presented and take it from there. We hope that you can take into consideration the comprehensive nature of the manuscript, along with the largely overwhelming support of the other four reviewers when reconsidering the manuscript.

Round 2

Reviewer 3 Report

The authors have successfully addressed all relevant issues.